# MODALITY-BALANCING PREFERENCE OPTIMIZATION OF LARGE MULTIMODAL MODELS BY ADVERSARIAL NEGATIVE MINING

## ABSTRACT

The task adaptation and alignment of Large Multimodal Models (LMMs) have been significantly advanced by instruction tuning and further strengthened by recent preference optimization. Yet, most LMMs still suffer from severe modality imbalance during reasoning, i.e., outweighing language prior biases over visual inputs, which bottlenecks their generalization to downstream tasks and causes hallucinations. However, existing preference optimization approaches for LMMs do not focus on restraining the internal biases of their Large Language Model (LLM) backbones when curating the training data. Moreover, they heavily rely on offline data and lack the capacity to explore diverse responses adaptive to dynamic distributional shifts during training. Meanwhile, Group Relative Policy Optimization (GRPO), a recent method using online-generated data and verified rewards to improve reasoning capabilities, remains largely underexplored in LMM alignment. In this paper, we propose a novel preference learning framework, Modality-Balancing Preference Optimization (`MBPO`), to address the modality imbalance in LMMs. `MBPO` constructs a more effective offline preference dataset by generating hard negatives, i.e., rejected responses misled by LLM biases due to limited usage of visual information, through adversarial perturbation of input images. Moreover, `MBPO` leverages the easy-to-verify nature of close-ended tasks to generate online responses with verified rewards. GRPO is then employed to train the model with offline-online hybrid data. Extensive experiments demonstrate that `MBPO` can enhance LMM performance on challenging vision-language tasks and effectively reduce hallucinations.

## 1 INTRODUCTION

*Large Multimodal Models* (LMMs) have achieved incredible success by integrating vision models with pre-trained *Large Language Models* (LLMs) through instruction tuning, enabling effective adaptation to diverse visual tasks (Liu et al., 2023; 2024a; Bai et al., 2025; Tong et al., 2024; Chen et al., 2024e; 202, 2023; Xiong et al., 2024; Chen et al., 2023; 2024d). Despite their strong performance across complex visual understanding scenarios, LMMs still face several fundamental challenges: achieving proper alignment between multimodal inputs (Li et al., 2024; Liu et al., 2023); collecting and effectively leveraging high-quality aligned multimodal data with accurate annotations (Tong et al., 2024; Luo et al., 2024); and mitigating hallucination, where models generate content disconnected from or contradicting the visual evidence (Yu et al., 2024a; Zhao et al., 2023). Furthermore, recent studies show that LMMs suffer from the modality imbalance problem, tending to over-rely on their language backbone while underutilizing the rich information available in visual inputs (Liu et al., 2024b; Jiang et al., 2024), thus leading to problematic behaviors such as incorrect visual perception and hallucinated responses.

To further improve task adaptation and alignment with human intent, recent studies (Yu et al., 2024a; Zhou et al., 2024b; Lu et al., 2025) adopt preference learning as a post-training strategy for LMMs, enhancing performance in general vision-language tasks and reducing hallucination. Benefiting from the simplified reward parameterization introduced by *Direct Preference Optimization* (DPO) (Rafailov et al., 2023), some works (Yu et al., 2024a; Pi et al., 2024; Cui et al., 2024; Jiang et al., 2024; Yu et al., 2024b; Amirloo et al., 2024) propose various strategies for constructing

pairwise preference datasets, typically selecting high-quality responses as preferred examples and hallucinated ones as rejected. While these methods help align model outputs with human preferences, they do not explicitly tackle the modality imbalance issue—where LMMs tend to over-rely on the linguistic priors of the language backbone rather than grounding their predictions in visual input. Furthermore, the inherently offline nature of DPO—relying exclusively on pre-collected model responses—limits its ability to adapt to distributional shifts during training, thereby hindering optimization effectiveness (Chen et al., 2024f;a). In contrast, the recently proposed *Group Relative Policy Optimization* (GRPO) (Shao et al., 2024) improves reasoning capabilities by utilizing online model-generated trajectories with verifiable reward signals (Guo et al., 2025). Recent studies (Chen et al., 2025; Shen et al., 2025; Zheng et al., 2025) have explored the potential of using GRPO to visual reasoning tasks, such as multimodal math problems and visual perception. However, the broader potential of reinforcement learning with verified rewards for general multimodal alignment remains largely underexplored.

In this paper, we propose Modality-Balancing Preference Optimization (`MBPO`), a novel framework that combines both offline and online preference data to address modality imbalance and improve alignment in LMMs. This framework comprises two complementary components: (1) an offline pairwise preference dataset constructed using adversarially mined negative responses, and (2) an online dataset with verifiable rewards collected dynamically during training.

- For the offline dataset, we focus on addressing modality imbalance issue, where the model tends to rely more on the language backbone's prior knowledge than on visual evidence. We first introduce an image information gain metric that quantifies how much visual content is utilized in a response. To generate rejected responses with low image information gain and high modality imbalance, we apply adversarial perturbations to the input image to reduce the model's confidence in the original ground-truth response. The perturbed image is then used, together with the original instruction, to produce a less visually grounded rejected response.

- For the online dataset, we leverage closed-ended visual instruction-tuning data (i.e., multiple-choice and yes/no questions) with verifiable answers. During training, the model generates multiple candidate responses for each input instruction, and rewards are assigned based on factual correctness. To avoid generating extremely short responses, we add a simple prompt instruction and an extra format reward to the online dataset. By adapting to distributional shifts throughout training, these reward signals enable more effective model alignment.

We jointly optimize the model using both offline and online data through the Group Relative Policy Optimization (GRPO) objective. Experimental results on a wide range of vision language tasks and hallucination benchmarks demonstrate that `MBPO` significantly mitigates modality imbalance and enhances overall performance.

Overall, **our contributions** can be summarized as follows:

- We propose `MBPO`, a novel framework that addresses modality imbalance in large multimodal models (LMMs) to improve alignment. By mining adversarial images to construct rejected responses, `MBPO` explicitly incentivizes LMMs to incorporate visual information during response generation.

- We leverage the easy-to-verify nature of close-ended data as an online dataset and use a simple prompt instruction along with a format reward to encourage the model to generate more diverse responses, including verifiable single-word answers and corresponding explanations.

- Experiments across general vision-language tasks and hallucination benchmarks demonstrate that `MBPO` effectively enhance LMM performance while effectively mitigating modality imbalance.

## 2 RELATED WORK

**Multimodal Preference Learning**. Preference learning is a proven method to align pretrained LLMs (Ouyang et al., 2022; McAleese et al., 2024) and LMMs (Sun et al., 2023) with human intentions and reduce model hallucination. Specifically, *Direct Preference Optimization* (DPO) (Rafailov et al., 2023) has been widely adopted for its elimination of an explicit reward model, enabling direct

optimization over pairs of preferred and rejected responses. Prior works have collected multimodal preference datasets using human annotations (Yu et al., 2024a) or AI-generated feedback (Li et al., 2023; Xiong et al., 2024). Another line of papers focus on self-rewarding (Yuan et al., 2024; Chen et al., 2024f) mechanisms, gathering preference data from model-generated response without external supervision. These approaches typically involve the design of evaluation prompts (Wang et al., 2024c), sentence-level search strategies (Zhou et al., 2024b) or decomposition into fine-grained judgments (Yu et al., 2024b; Cui et al., 2024). Although some methods re-collect preference data for multi-round iterative training, the inherently offline nature of DPO leads them to rely heavily on pre-collected model responses within each epoch, making it difficult to adapt to distribution shifts during training. In contrast, our method combines online and offline samples for both dynamic and consistent preference alignment.

**Noise Injection in Multimodal Preference Learning** While human annotations are costly and AI-generated feedback is susceptible to reward hacking (Skalse et al., 2022) and lacks verifiability, some studies create rejected responses by deliberate error injections. some works (Pi et al., 2024; Zhou et al., 2024a) apply Gaussian distortions to input images and employ LLM or LMM to introduce hallucinated responses, while Wang et al. (2024a) apply random cropping on images. More recently, Liu et al. (2025a) use distorted image inputs in GRPO training to enhance LMM reasoning in multimodal math. However, rejected responses generated with random image distortion or external rewriting may not yield clearly incorrect outputs and often lie far from the model generation distribution. Our work focuses on adversarial inputs that produce in-domain, instruction-following responses that are incorrect yet highly probable under the model's distribution.

**Multimodal RLVR.** Recent studies show that large-scale reinforcement learning significantly enhances LLM in complex reasoning (Jaech et al., 2024; Guo et al., 2025; Team et al., 2025). Several concurrent works extend *Reinforcement Learning with Verifiable Rewards* (RLVR), as used in Deepseek-R1 to multimodal settings. One line of research focuses on multimodal math (Meng et al., 2025; Huang et al., 2025), academic questions (Peng et al., 2025; Yang et al., 2025), while others target visual perception tasks (Yu et al., 2025) such as counting Chen et al. (2025), grounding (Shen et al., 2025), detection (Zhan et al., 2025), and refering segmentation (Liu et al., 2025b). In our paper, we extend RLVR to broader visual domains, including general visual question answering, open-ended visual chat and hallucination related tasks.

## 3 PRELIMINARIES

**Adversarial Attacks** on images can mislead LMMs into generating incorrect or misleading responses. To expose worst-case vulnerabilities of the model, adversarial images can be crafted by *Projected Gradient Descent* (PGD) (Madry et al., 2017), the multistep extension of the *Fast Gradient Sign Method* (FGSM) (Goodfellow et al., 2014) that is widely regarded as the strongest first-order $\ell_\infty$ attack. Beginning from either the clean input $x$ or a random point $x^{(0)} \sim \mathcal{U}(x - \epsilon, x + \epsilon)$ inside the $\ell_\infty$ ball of radius $\epsilon$, PGD perform $T$ iterative updates

$$x^{(t+1)} = \Pi_{B_\epsilon(x)}\Big(x^{(t)} + \alpha \cdot \text{sign}\big(\nabla_x J(\theta, x^{(t)}, y)\big)\Big), \qquad t = 0, \ldots, T-1, \tag{1}$$

where $\alpha$ is the step size, $\theta$ is the parameter of model and $J(\theta, x, y)$ is the loss, and $\Pi_{B_\epsilon(x)}(\cdot)$ projects its argument back onto the $\ell_\infty$ ball $B_\epsilon(x) = \{\tilde{x} : \|\tilde{x} - x\|_\infty \leq \epsilon\}$. After the final iteration, PGD clips $x^{(T)}$ to the valid data range to obtain the adversarial example $x^{\text{adv}}$. By following the steepest ascent direction at each step yet remaining within the prescribed perturbation budget, PGD yields perturbations that are imperceptible to humans but significantly degrade model performance, providing a stringent evaluation of robustness.

**Group Relative Policy Optimization (GRPO)** (Shao et al., 2024; Guo et al., 2025) has been proven effective on LLMs. Instead of relying on a critic model, which is typically as large as the policy model, this approach estimates the baseline using group scores. Specifically, for each question $q$, GRPO samples a set of outputs $\{o_1, o_2, \ldots, o_G\}$ from the old policy $\pi_{\theta_{\text{old}}}$, and then updates the policy model $\pi_\theta$ by maximizing the following objective:

$$\mathcal{J}_{GRPO}(\theta) = \mathbb{E}[q \sim P(Q), \{o_i\}_{i=1}^G \sim \pi_{\theta_{\text{old}}}(O \mid q)]$$

$$\frac{1}{G} \sum_{i=1}^G \Big\{ \min\Big( \frac{\pi_\theta(o_i \mid q)}{\pi_{\theta_{\text{old}}}(o_i \mid q)} A_i, \ \text{clip}\Big( \frac{\pi_\theta(o_i \mid q)}{\pi_{\theta_{\text{old}}}(o_i \mid q)}, 1 - \epsilon, 1 + \epsilon \Big) A_i \Big) \tag{2}$$

$$- \ \beta \, \mathbb{D}_{KL}\big(\pi_\theta \, \| \, \pi_{\text{ref}}\big) \Big\},$$

where $\epsilon$ and $\beta$ are hyperparameters, and $A_i$ denotes the advantage, which is computed based on a group of rewards $\{r_1, r_2, \ldots, r_G\}$ associated with the outputs in each group:

$$A_i = \frac{r_i - \text{mean}(\{r_1, r_2, \cdots, r_G\})}{\text{std}(\{r_1, r_2, \cdots, r_G\})}. \tag{3}$$

To prevent the updated policy $\pi_\theta$ from deviating too far from the stable reference $\pi_{\text{ref}}$, GRPO loss has a *Kullback-Leibler Divergence* term $\mathbb{D}_{KL}$ which is estimated with an unbiased estimator:

$$\mathbb{D}_{KL}\left(\pi_\theta || \pi_{ref}\right) = \frac{\pi_{ref}(o_i|q)}{\pi_\theta(o_i|q)} - \log \frac{\pi_{ref}(o_i|q)}{\pi_\theta(o_i|q)} - 1, \tag{4}$$

## 4 METHODOLOGY

MBPO is a hybrid preference learning framework designed to enhance alignment and mitigate the modality imbalance problem in LMMs. It combines both offline and online preference data to provide stable yet adaptive reward signals throughout training. Section 4.1 introduces how MBPO constructs the offline preference dataset, where the chosen responses are accurate and visually grounded, and the rejected responses rely heavily on the LLM backbone's prior knowledge, neglecting visual information. These modality-imbalanced rejected responses are generated by adding adversarial noise to input images, which suppresses visual cues and triggers the prior biases from the LLM backbone. Section 4.2 describes how MBPO performs online exploration using closed-ended data with verifiable rewards. With a simple prompt instruction and an extra format reward, MBPO enhances the model's ability to explore diverse responses and dynamically adapt to distributional shifts during training. An overview of our training pipeline is illustrated in Figure 1.

### 4.1 OFFLINE PREFERENCE DATA CONSTRUCTION

Current LMMs often suffer from the modality imbalance problem that model responses overweigh the prior biases of the LLM backbone and underutilize the visual information from the image encoder, leading to incorrect or insufficient visual content in the output responses. To address this issue, MBPO is designed to balance different input modalities to incorporate more accurate and relevant visual information into the the model responses. To quantify this, we propose a metric called *Image Information Gain* (IIG), which measures the amount of visual information contained in the generated response. Given data consisting of a question $q$, an image $I$, and a response $o$, IIG is defined as:

$$\text{IIG}(o, q, I) = - \log p_\theta(o \mid q, I_b) + \log p_\theta(o \mid q, I) \tag{5}$$

where $I_b$ denotes a blank image (all-zero pixels) of the same dimensions as $I$. This metric captures the difference in output probability when conditioned on the actual image versus a blank image with no information, using the same question and response. A larger IIG value indicates that the response $o$ incorporates more information from the image $I$. As the goal of MBPO is to encourage LMMs to incorporate more visual information into their responses, we select data whose responses have high IIG scores from a visual instruction tuning dataset as our preference dataset and chosen responses.

The next step is to construct the corresponding rejected responses for the selected data. Compared to the chosen responses that contain rich image information, the rejected responses should include limited visual information and rely primarily on the prior biases of the LLM backbone. To generate

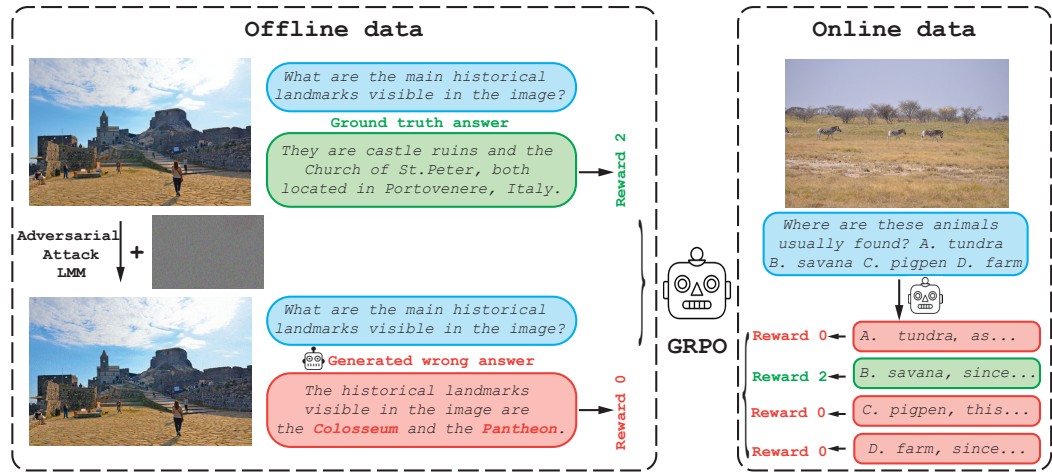

Figure 1: Overview of MBPO framework. To construct the offline preference dataset, we generate adversarial perturbations for each input image to minimize the output probability of the chosen response. Rejected responses are then generated using these adversarially perturbed images. This process amplifies modality imbalance, causing the LMM to rely more heavily on the prior biases of its LLM backbone rather than the visual information. In parallel, MBPO incorporates an online dataset composed of closed-ended examples, where response correctness can be easily verified. During training, the LMM generates multiple responses, and verified rewards are assigned based on their correctness. Finally, the offline and online datasets are combined to optimize the LMM using the MBPO loss in a hybrid training paradigm.

the rejected responses, MBPO adds adversarial noise to the image to minimize the output probability of the chosen responses:

$$I^{(t+1)} = \Pi_{B_\epsilon(\mathbf{I})} \left( I^{(t)} + \alpha \cdot \text{sign} \left( \nabla_I \left( -\log p_\theta(o_w \mid q, I) \right) \right) \right), \qquad t = 0, \ldots, T-1 \qquad (6)$$

where $o_w$ is the winner/chosen response from the visual instruction dataset. $I^0$ is the original image from the visual instruction dataset, and we denote the final $I^T$ as the adversarial image $I_{\text{adv}}$. After obtaining the adversarial image, we sample a loser/rejected response using the same question $q$:

$$o_l \sim \pi_\theta(\cdot \mid q, I_{\text{adv}}) \qquad (7)$$

As the adversarial image is perturbed to minimize the output probability of the correct chosen response, it loses visual information relevant to that response. When the model generates a new response using the adversarial image, it cannot effectively retrieve the visual information from the image and instead relies on the prior biases of the LLM backbone.

The chosen responses from the visual instruction dataset and the generated rejected responses using adversarial images constitute our offline preference dataset: $D_{\text{offline}} = \{(q, I, o_w, o_l)\}$. During our training, we assign hard rewards to the offline data. Specifically, a reward of 2 is given to the chosen response, and a reward of 0 is assigned to the rejected response.

## 4.2 ONLINE PREFERENCE DATA

Although training on offline datasets can improve a model's performance, they still face several limitations. First, they cannot adapt to the latest distribution shifts during training, limiting their training effectiveness on the offline data (Chen et al., 2024f). Moreover, offline data typically consists of pairwise preference annotations, which represent only a limited set of possible model responses. In contrast, online preference learning methods (Peng et al., 2025; Guo et al., 2025) generate multiple responses using the latest model weights, allowing optimization over the current output distribution and enabling the sampling of more possible responses. Furthermore, they can provide accurate feedback to online generations using verified rewards, rather than relying on unreliable reward models or costly human verification.

In the visual instruction dataset, we observe that the responses for multiple-choice and yes/no questions are easy to verify using verifiable checking (Shao et al., 2024). Therefore, we construct our online preference data using all the multiple-choice and yes/no samples from `MMSeed`, totaling around $2k$ examples. However, in the original visual instruction dataset, the multiple-choice data are prompted with `"Answer with the option's letter from the given choices directly."`, which results in responses with limited diversity, restricted to just a few option letters. Therefore, we replace it with a new simple prompt: `"Answer with the option's letter from the given choices first, and only after that, provide a detailed explanation for the choice."`.

For each sample, the model generates multiple responses using random decoding. The correctness of each response is verified by matching it with the ground-truth answer—either the correct option letter or the "yes"/"no" word. A reward of 2 is assigned to correct responses, while incorrect responses receive a reward of 0. Furthermore, to ensure that the model follows the instructions and provides diverse responses for both multiple-choice and yes/no data, we add an extra format reward to the online data: if a response contains fewer than $\tau$ words, we apply a $\gamma$ penalty to the reward:

$$r_i = 2 \cdot \mathbf{1}_{\hat{y}_i = y_i} - \gamma \cdot \mathbf{1}_{L_i < \tau} \tag{8}$$

where $\hat{y}_i$ denotes the correct letter, $y_i$ denotes the generated letter, and $L_i$ denotes the number of words in the response. $\gamma$ and $\tau$ are two hyperparameters. In this way, we encourage the model to provide an explanation after the verifiable option letter, rather than generating only a single option letter.

Overall, to exploit the complementary strengths of both online and offline preference data, `MBPO` integrates them into a unified hybrid preference dataset. During training, `MBPO` randomly samples mini-batches from this combined dataset. For samples coming from the offline dataset, rewards are directly assigned to the chosen and rejected responses based on the known preference. For samples drawn from the online dataset, `MBPO` first generates multiple candidate responses using the current policy model $\pi_\theta$, and then assigns rewards according to their agreement with the ground truth answer.

## 5 EXPERIMENTS

In this section, we first introduce the implementation details, including training details, datasets, evaluation protocol and baseline methods. Subsequently, we present our main results comparing `MBPO` with baseline methods on several general vision language tasks and hallucination benchmarks, demonstrating the effectiveness of `MBPO`. In addition, the ablation study provides a closer look at `MBPO` and verifies the contributions of its individual components. Lastly, we include additional experimental results for further analysis.

### 5.1 IMPLEMENTATION DETAILS

**Training details:** Following recent studies (Shen et al., 2025; Chen et al., 2025; Zheng et al., 2025) that apply GRPO to train LMMs, we adopt `Qwen2-VL-7B-Instruct` (Wang et al., 2024b) and `Qwen2.5-VL-7B-Instruct` (Bai et al., 2025) as our backbone models. The learning rate is set to $5 \times 10^{-7}$, and the KL-divergence coefficient ($\beta$) is set to 0.1. Gradient accumulation is used to maintain an effective batch size of 16. For each multiple-choice and yes/no sample, we generate 16 responses to compute the GRPO advantage. A reward of 2 is assigned to correct responses, and 0 otherwise. $\gamma$ and $\tau$ are set to 0.5 and 5 respectively. For offline data, chosen responses are assigned a reward of 2, while rejected responses receive reward 0. To enable efficient training, we use `bfloat16` precision. For the adversarial image generation, we attack each image 20 iterations and the step size $\alpha$ is set as $\frac{4}{255}$. All experiments are conducted using `PyTorch` and the Hugging Face `Transformers` library on $4\times$ NVIDIA H100 80GB GPUs.

**Datasets:** Following previous works (Pi et al., 2024; Zhou et al., 2024a), we use high-quality visual instruction tuning data as our offline positive samples to train the powerful and up-to-date Qwen series models. Specifically, from the high-quality `MMSeed-163K` dataset (Luo et al., 2024), we randomly select $10K$ samples with high IIG for the offline dataset, along with all

multiple-choice and yes/no samples (approximately 2K) as the online dataset. The `MMSeed-163K` dataset is a diverse multi-domain instruction dataset curated from LLaVA-Instruct (Liu et al., 2023), ShareGPT4V (Chen et al., 2024b), and Cambrian-1 (Tong et al., 2024), encompassing 163K samples across tasks such as VQA, OCR, chart understanding and reasoning. More details can be found in the Appendix.

**Evaluation protocol:** We conduct a wide range of benchmarks to evaluate the comprehensive capabilities of LMMs, covering both general vision language tasks and hallucination benchmarks. For general vision language tasks, we use AI2D (Kembhavi et al., 2016), MME (Fu et al., 2023), MMStar (Chen et al., 2024c), MMVet (Yu et al., 2024c) and MMBench (Liu et al., 2024c). For hallucination benchmarks, we use MMHal-Bench (Sun et al., 2023) and ObjectHal (Rohrbach et al., 2018). The evaluation is performed using the popular `LMMs-Eval` framework (Zhang et al., 2024). More details about these benchmarks can be found in the Appendix.

**Baselines:** We select studies that use preference learning to align LMMs as our baselines, including BPO (Pi et al., 2024), POVID (Zhou et al., 2024a), RLAIFV (Yu et al., 2024b), SIMA (Wang et al., 2024c), CSR (Zhou et al., 2024b), mDPO (Wang et al., 2024a), MFPO (Jiang et al., 2024), FiSAO (Cui et al., 2024), and DAMA (Lu et al., 2025). For BPO, POVID, RLAIF-V, and CSR, we download their publicly released model weights and report evaluation results with the `LMMs-Eval` framework. For other methods, we report the results of the 7B model reported in their original papers. To ensure a fair comparison, we also train `Qwen2/2.5-VL-7B-Instruct` on the corresponding public datasets from BPO, POVID, RLAIF-V and CSR as additional baselines. More details about the baselines are provided in the Appendix.

## 5.2 BENCHMARK COMPARISONS

In this section, we compare the performance of baseline methods and `MBPO` on general vision-language tasks and hallucination benchmarks. The detailed results are presented in Table 1. If a baseline model is not available or the original paper does not report results on a specific benchmark, we use a "–" in the table. On general vision-language tasks such as $MME^P$, MMStar, and MMVet, `MBPO` consistently outperforms all baselines with both Qwen base models. For example, `MBPO` using `Qwen2-VL-7B` surpasses the second-best result on $MME^P$ by 5.7 points and on MMVet by 1.9 points. When using the `Qwen2.5-VL-7B` backbone, `MBPO` improves MMStar performance from 62.0 to 63.0, and MMVet from 62.2 to 65.8. On the AI2D benchmark, which evaluates the factual knowledge of LMMs, all methods, including `MBPO`, perform similarly and do not show significant improvements. This suggests that preference learning strategies cannot effectively enhance the factual knowledge of LMMs. On hallucination benchmarks MMHal-Bench and ObjectHal, `MBPO` achieves the best performance across most metrics. With `Qwen2-VL-7B`, `MBPO` reduces $CHAIR_S$ and $CHAIR_I$ by 3.3 and 1.6 points respectively, compared to the base model. These reductions are even more pronounced with `Qwen2.5-VL-7B`, where $CHAIR_S$ drops from 14.1 to 7.4, and $CHAIR_I$ from 6.9 to 3.6, nearly halving the hallucination error. In addition, `MBPO` improves $MMHal^{score}$ from 3.68 to 3.75 and reduces $MMHal^{rate}$ from 0.42 to 0.34, indicating fewer hallucinations in model responses. In summary, `MBPO` yields consistent and superior performance across a wide range of benchmarks based on the average of scores. It not only improves results on general vision-language tasks, but also significantly alleviates hallucination. These results highlight the advantage of encouraging LMMs to rely more on input visual information rather than the prior biases of the LLM backbone.

## 5.3 ABLATION STUDY

We conduct an ablation study on two Qwen base models across both general vision language tasks and hallucination benchmarks, following the same implementation details described in Section 5.1. To evaluate the effectiveness of each component in `MBPO`, we incrementally add each one to the framework and measure its impact on each benchmark. The results are shown in Table 2, where +*offline rand.* denotes offline rejected responses constructed using random noise sampled from $\mathcal{N}(0,1)$. +*offline adv.* indicates using only our offline dataset for training, and +*online* refers to training the model solely on our online dataset. Based on the results, `MBPO` achieves the best performance on 7 out of 10 benchmarks with `Qwen2-VL-7B` and on 6 out of 10 benchmarks with `Qwen2.5-VL-7B`. Furthermore, `MBPO` performs the second best on 2 of 10 benchmarks with

Table 1: Comparison with baseline methods on general vision language and hallucination benchmarks. $^*$ indicates results reported in the original papers, and ↓ indicates that lower is better. The best performance is marked in **bold**.

| Model | AI2D | MME$^p$ | MMStar | MMVet | MMB | MMHal$^{score}$ | Avg | MMHal$^{rate}$↓ | CHAIR$_S$↓ | CHAIR$_I$↓ | Avg↓ |
|---|---|---|---|---|---|---|---|---|---|---|---|
| BPO | – | – | – | 36.8$^*$ | – | – | – | – | 31.9$^*$ | 15.1$^*$ | – |
| POVID | 54.2 | 1438.7 | 35.6 | 31.9 | 64.3 | 2.1 | 1626.8 | 0.60 | 37.9 | 18.9 | 57.4 |
| RLAIFV | 52.3 | 1356.0 | – | 24.0 | 62.7 | 2.9 | – | 0.46 | 8.6 | 4.3 | 13.4 |
| SIMA | – | 1507.7$^*$ | – | 31.6$^*$ | 64.9$^*$ | 2.3$^*$ | – | – | 40.9$^*$ | 10.4$^*$ | – |
| CSR | 54.9 | 1523.3 | 34.3 | 31.1 | 64.1 | 2.2 | 1709.9 | 0.6 | 12.2 | 8.3 | 21.1 |
| mDPO | – | – | – | – | – | 2.39$^*$ | – | 0.54$^*$ | 35.7$^*$ | 9.8$^*$ | 46.1 |
| MFPO | – | – | – | – | – | 2.89$^*$ | – | 0.45$^*$ | 10.6$^*$ | 5.1$^*$ | 16.2 |
| FiSAO | – | 1522.6$^*$ | – | 30.7$^*$ | 64.8$^*$ | – | – | – | 39.9$^*$ | 9.9$^*$ | – |
| DMMA | – | – | – | 32.8$^*$ | – | 2.76$^*$ | – | 0.41$^*$ | – | – | – |
| Qwen2-VL-7B | 80.4 | 1692.7 | 57.1 | 57.9 | 78.9 | 3.50 | 1970.5 | 0.34 | 10.9 | 5.9 | 17.1 |
| +BPO | **80.6** | 1684.3 | 57.0 | 58.4 | 79.2 | 3.55 | 1963.1 | 0.31 | 8.7 | 4.8 | 13.8 |
| +POVID | **80.6** | 1690.2 | **57.6** | 58.9 | 78.6 | 3.53 | 1969.4 | **0.29** | 11.6 | 7.2 | 19.1 |
| +RLAIF-V | 80.4 | 1696.4 | 57.1 | 56.9 | 78.1 | 3.38 | 1972.3 | 0.34 | 9.2 | 5.6 | 15.1 |
| +CSR | **80.6** | 1697.1 | 57.1 | 57.0 | 78.5 | 3.38 | 1973.7 | 0.35 | 21.4 | 11.6 | 33.4 |
| +MBPO (ours) | **80.6** | **1702.8** | **57.6** | **60.8** | **79.4** | **3.58** | **1984.5** | 0.36 | **7.6** | **4.3** | **12.3** |
| Qwen2.5-VL-7B | 82.6 | 1680.1 | 62.0 | 62.2 | 83.2 | 3.68 | 1973.8 | 0.42 | 14.1 | 6.9 | 21.4 |
| +BPO | **82.7** | 1659.8 | 62.9 | 63.7 | 83.5 | 3.51 | 1956.1 | 0.42 | 9.9 | 5.4 | 15.7 |
| +POVID | 82.6 | 1669.1 | 62.6 | 63.8 | 83.5 | 3.73 | 1965.3 | 0.37 | 10.5 | 5.7 | 16.6 |
| +RLAIF-V | **82.7** | 1686.3 | 62.7 | 63.8 | **83.6** | 3.63 | 1982.7 | 0.41 | 11.8 | 6.4 | 18.6 |
| +CSR | 82.6 | 1687.8 | 62.1 | 61.7 | **83.6** | 3.71 | 1981.5 | 0.41 | 18.3 | 11.0 | 29.7 |
| +MBPO (ours) | 82.5 | **1706.3** | **63.0** | **65.8** | **83.6** | **3.75** | **2005.0** | **0.34** | **7.4** | **3.6** | **11.3** |

Table 2: Ablation studies of adding each component of MBPO and their results on general vision language and hallucination benchmarks. +*offline,rand.* indicates that the offline rejected samples are generated using images with random noise. We mark the best performance in **bold**.

| Model | AI2D | MME$^p$ | MMStar | MMVet | MMB | MMHal$^{score}$ | Avg | MMHal$^{rate}$↓ | CHAIR$_S$↓ | CHAIR$_I$↓ | Avg↓ |
|---|---|---|---|---|---|---|---|---|---|---|---|
| Qwen2-VL-7B | 80.4 | 1692.7 | 57.1 | 57.9 | 78.9 | 3.50 | 1970.5 | 0.34 | 10.9 | 5.9 | 17.1 |
| +offline, rand. | **80.6** | 1684.8 | 57.8 | 58.5 | 78.6 | 3.54 | 1963.8 | 0.36 | 10.2 | 5.3 | 15.9 |
| +offline, adv. | 80.5 | 1697.6 | **58.0** | 59.8 | 78.8 | 3.50 | 1978.2 | **0.33** | 7.8 | 4.8 | 12.9 |
| +online, $\gamma = 0$ | 80.5 | 1682.3 | 57.5 | 59.0 | 78.4 | 3.46 | 1961.1 | 0.36 | 8.4 | 4.5 | 13.3 |
| +online, $\gamma = 0.5$ | 80.5 | 1681.9 | 57.4 | 60.6 | 78.4 | 3.52 | 1962.3 | 0.35 | 8.3 | **4.3** | 12.9 |
| +MBPO | **80.6** | **1702.8** | 57.6 | **60.8** | **79.4** | **3.58** | **1984.5** | 0.36 | **7.6** | **4.3** | **12.3** |
| Qwen2.5-VL-7B | 82.6 | 1680.1 | 62.0 | 62.2 | 83.2 | 3.68 | 1973.8 | 0.42 | 14.1 | 6.9 | 21.4 |
| +offline, rand. | 82.6 | 1688.4 | 62.1 | 61.5 | 83.4 | 3.57 | 1981.6 | 0.38 | 12.4 | 6.0 | 18.8 |
| +offline, adv. | 82.5 | 1693.5 | 62.1 | 61.3 | 83.3 | 3.56 | 1986.3 | 0.38 | 7.5 | 4.1 | 12.0 |
| +online, $\gamma = 0$ | **82.8** | 1693.8 | 62.8 | 64.3 | 83.5 | 3.83 | 1991.0 | 0.39 | 10.6 | 6.1 | 17.1 |
| +online, $\gamma = 0.5$ | **82.8** | 1693.6 | 62.9 | 64.7 | **83.6** | **3.91** | 1991.5 | 0.37 | 9.8 | 5.4 | 15.6 |
| +MBPO | 82.5 | **1706.3** | **63.0** | **65.8** | **83.6** | 3.75 | **2005.0** | **0.34** | **7.4** | **3.6** | **11.3** |

Qwen2.5-VL-7B. Thus, we conclude that each component of MBPO is essential for achieving the best performance in most cases.

## 5.4 FURTHER ANALYSIS

**Adversarial Image:** To gain a deeper understanding of the adversarial image, we present a detailed case study comparing model responses on an image with adversarial noise versus an image with random noise. The images and corresponding responses from Qwen2-VL-7B and Qwen2.5-VL-7B are shown in Fig. 2. For the adversarial image, both models follow the question's instruction and provide responses based on the prior biases of the LLM backbone. In contrast, for the image

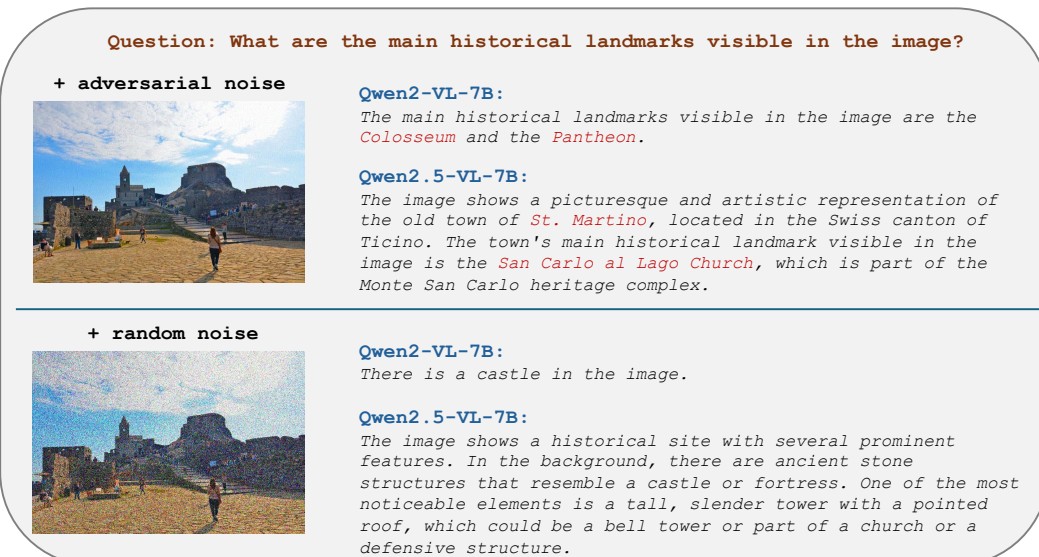

Figure 2: An example comparing model responses of the image with adversarial noise and random noise. The prior bias from LLM is marked in red.

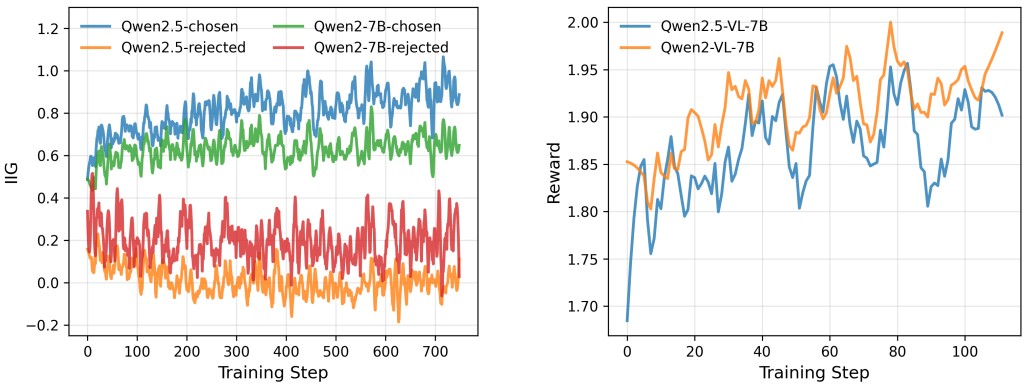

Figure 3: IIG of chosen and rejected responses change along with the training.

Figure 4: Reward of the online closed-end data changes along with the training.

with random noise, both models fail to follow the question's instruction and instead offer a general description of the image, without leveraging the LLM's prior biases.

**Image Information Gain:** The goal of `MBPO` is to encourage LMMs to extract more information from the image, reflected by a higher IIG after training. Using the same +*offline adv.* setting described in Section 5.3, we train the model on offline dataset and measure the change in IIG during training. The smoothed results are shown in Fig 3. As illustrated, the IIG of chosen responses increases throughout the training process, while the IIG of rejected responses remains consistently low. These results demonstrate that `MBPO` effectively addresses the modality imbalance problem and successfully incorporates more visual information into the responses by training on our offline preference data.

**Closed-set Data Reward:** To verify the effectiveness of learning from the online preference dataset, we measure the reward on closed-end data during training. The settings follow those of the +*online* configuration in Section 5.3, where each model is trained on online closed-end data for one epoch. As shown in the smoothed results in Fig. 4, the reward of closed-end data increases as training progresses for all models. This demonstrates the effectiveness of our online learning strategy, which improves model performance on closed-end questions through GRPO training.

## ETHICS STATEMENT

Our approach enhances LMM alignment by explicitly addressing modality imbalance, encouraging models to rely more on visual inputs rather than language priors. Technically, `MBPO` offers a new perspective on leveraging both adversarially generated offline data and online verified responses for training, which may inspire future research in multimodal alignment. Our method helps reduce hallucinations in LMMs, a key challenge for deploying such models in real-world applications. While `MBPO` significantly improves factual grounding, hallucinations can still occur. Therefore, we emphasize the importance of safety measures, robust evaluation, and responsible deployment when applying this method in practice.

## REPRODUCIBILITY STATEMENT

We describe the implementation details in Section 5.1 to ensure clarity and reproducibility. More-over, the complete source code is included in the supplementary materials, allowing readers to re-produce all experiments and results presented in this work.

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

# Appendix

## A  EXPERIEMNTAL DETAILS

### A.1  DATASET

The `MMSeed-163K` dataset (Luo et al., 2024) is a curated collection of $163K$ high-quality image-text instruction samples designed to support multimodal language model training. It integrates and refines data from `LLaVA-Instruct` (Liu et al., 2023), `ShareGPT4V` (Chen et al., 2024b), and `Cambrain-1` (Tong et al., 2024), covering diverse instruction formats including dialogue-based QA, global descriptions, scientific reasoning, and chart interpretation. As the goal of `MBPO` is to balance modality in *Large Multimodal Models* (LMMs) by encouraging more visual information to be used, we use responses with high IIG as the chosen responses in our offline dataset. To construct our offline dataset efficiently, we first randomly select $60K$ samples without closed-end questions from the `MMSeed-163K`. Then we use `Qwen2-VL-2B` (Wang et al., 2024b) to compute the *Image Information Gain* (IIG) of each sample and choose $10K$ samples with the highest IIG as our offline dataset.

### A.2  EVALUATION BENCHMARKS

- **AI2D** (Kembhavi et al., 2016) is a large-scale dataset designed to evaluate a model's ability to interpret and reason about grade school science diagrams. It contains over 5,000 annotated diagrams with more than 150,000 detailed annotations, syntactic parses, and 15,000+ multiple-choice questions. The benchmark focuses on two key tasks: Syntactic Parsing, which involves detecting diagram components and their structural relationships, and Semantic Interpretation, which maps these components to real-world concepts and events.

- **MME** (Fu et al., 2023) is a comprehensive benchmark designed to evaluate LMMs across two core dimensions: *perception* (MME$^P$) and *cognition* (MME$^c$). It consists of 14 subtasks, each crafted to assess a model's ability to interpret visual content and reason about it. For each image, the benchmark poses two questions whose answers are marked yes [Y] and no [N], respectively, allowing for a fine-grained evaluation of LMMs.

- **MMStar** (Chen et al., 2024c) is a high-quality vision-indispensable benchmark designed to rigorously evaluate the multimodal capabilities of LMMs. It comprises 1,500 human-curated samples across 6 core capabilities and 18 fine-grained evaluation axes, offering a comprehensive and balanced assessment of models' understanding of both visual and textual modalities.

- **MMVet** (Yu et al., 2024c) is a comprehensive benchmark designed to evaluate the integration capabilities of generalist vision-language models. It defines six core VL abilities and systematically examines sixteen meaningful pairwise combinations to assess how well models can jointly reason over multiple modalities. To address the challenge of evaluating open-ended outputs, MMVet introduces an LLM-based evaluator. Specifically, we use the OpenAI API *gpt-4o-2024-08-06* as our evaluator model.

- **MMBench** (Liu et al., 2024c) is a comprehensive benchmark designed to objectively and systematically evaluate the capabilities of LMMs. It consists of over 3,000 multiple-choice questions spanning 20 ability dimensions, including object localization, social reasoning, and more. Each dimension includes approximately 125 questions, ensuring balanced coverage across various vision-language skills.

- **MMHal-Bench** (Sun et al., 2023) is a benchmark designed to evaluate hallucinations in large multimodal models (LMMs) through 96 adversarially constructed image-question pairs. These pairs span 8 hallucination types and cover 12 object topics from COCO. A GPT model (OpenAI *gpt-4o-2024-08-06*) is used as an evaluator by providing it with the image category, the question, the LMM's response, and a human-generated reference answer. The overall score and hallucination rate are reported to measure the model performance on MMHal-Bench.

- **ObjectHal** (Rohrbach et al., 2018) is a widely adopted benchmark for assessing common object hallucination in detailed image descriptions. Following Yu et al. (2024b), we employ 8 diverse prompts per image to improve evaluation stability. It assesses object hallucination at

the instance and sentence levels, which can be calculated as:

$$\text{CHAIR}_I = \frac{|\{\text{hallucinated objects}\}|}{|\{\text{all mentioned objects}\}|} \qquad \text{CHAIR}_S = \frac{|\{\text{captions with hallucinated objects}\}|}{|\{\text{all captions}\}|} \tag{9}$$

### A.3 BASELINES

- **BPO** (Pi et al., 2024) generates negative responses directly from the model to perform preference learning. It introduces two key strategies: (1) using distorted images to trigger language-biased outputs, and (2) using a text-only LLM to inject common but incorrect elements into otherwise correct responses. These bootstrapped negatives are paired with high-quality references to train the model via preference optimization.

- **POVID** (Zhou et al., 2024a) uses ground-truth instructions as preferred responses, and creates dispreferred responses through two different hallucination strategies: (1) prompting GPT-4V to inject plausible hallucinations into correct answers, and (2) distorting input images to elicit hallucinations from the VLM itself. These pairwise preference samples are then trained with Direct Preference Optimization (DPO).

- **RLAIF-V** (Yu et al., 2024b) introduces two key innovations to enhance reward learning from AI feedback. First, it improves feedback quality by generating candidate responses through multiple decoding trials under identical conditions, effectively removing confounding factors like text style. It also uses a divide-and-conquer strategy to break complex response evaluation into simpler claim-level judgments, enabling more accurate and efficient preference modeling. Second, for inference-time guidance, RLAIF-V employs a self-feedback mechanism using reward scores from models aligned via Direct Preference Optimization (DPO) to refine responses without external supervision.

- **SIMA** (Wang et al., 2024c) leverages existing vision instruction datasets to self-generate responses and uses an in-context self-critic mechanism to create preference pairs for tuning. By designing specialized critic prompts, SIMA enables the LMM itself to act as the judge, eliminating the need for extra fine-tuning. Additionally, it introduces three new visual metrics to guide the self-critique process, boosting the reliability of preference judgments.

- **CSR** (Zhou et al., 2024b) enables the model to refine itself by repeatedly generating candidate responses, scoring each with a reward function, and compiling the highest-rated examples into preference data for fine-tuning. In its reward-modeling phase, CSR follows a step-wise strategy and embeds visual constraints within the self-rewarding process to amplify the impact of visual signals.

- **mDPO** (Wang et al., 2024a) aligns LMMs by optimizing image preference data, rather than relying solely on text-based preference. To stabilize training, MDPO introduces a reward anchor that ensures chosen responses always receive positive rewards, mitigating the risk of degrading their likelihood.

- **MFPO** (Jiang et al., 2024) constructs image preference data by identifying hallucination-prone regions via keyword extraction and mapping them to image segments using the Segment Anything Model. Fine-grained noisy images are used as negative samples, and a reward function is built to favor clean over noisy regions. MFPO also incorporates a curriculum learning-inspired hierarchical alignment strategy that categorizes training data by difficulty (easy to hard), enabling stable and progressive learning. Margin loss is used to ensure consistent reward separation between preferred and rejected responses.

- **FiSAO** (Cui et al., 2024) is a self-alignment approach for LMMs that enhances multimodal alignment without requiring extra data. It leverages the model's own vision encoder as a fine-grained verifier to provide token-level feedback during training. This enables more precise supervision and improves alignment performance beyond traditional preference tuning methods.

- **DAMA** (Lu et al., 2025) dynamically adjusts the preference optimization coefficient $\beta$ based on both data hardness and the model's responsiveness. It measures the difficulty based on CLIP-based image-text similarity. Furthermore, it adapts $\beta$ based on real-time responsiveness inferred from reward gaps between preferred and rejected responses. This dual adaptation

Table 3: Exploration of the impact of iteration and step size in generating adversarial images for the offline dataset. $+(i,j)$ stands for $i$ iterations and $\frac{j}{255}$ step size. We mark the best performance **bold**.

| Model | AI2D | MME$^c$ | MME$^p$ | MMStar | MMVet | MMB | MMHal$^{score}$ | MMHal$^{rate}$ ↓ | CHAIR$_S$ ↓ | CHAIR$_I$ ↓ |
|---|---|---|---|---|---|---|---|---|---|---|
| Qwen2-VL-7B | 80.4 | 628.2 | 1692.7 | 57.1 | 57.9 | 78.9 | 3.50 | 0.34 | 10.9 | 5.9 |
| +(5,4) | 80.5 | 635.7 | 1704.9 | 57.5 | **60.0** | 78.9 | **3.57** | 0.40 | **4.5** | **2.5** |
| +(10,4) | **80.7** | 637.8 | 1701.7 | 57.9 | 57.5 | 79.0 | 3.54 | 0.40 | 6.4 | 3.5 |
| +(20,2) | 80.5 | **640.0** | **1706.5** | 57.7 | 59.4 | 78.8 | **3.57** | 0.39 | 7.6 | 4.0 |
| +(20,4) | 80.5 | 635.7 | 1697.6 | **58.0** | 59.8 | 78.8 | 3.50 | **0.33** | 7.8 | 4.8 |
| +(20,8) | **80.7** | 628.2 | 1700.4 | 57.7 | 59.3 | **79.3** | 3.39 | 0.41 | 7.0 | 3.9 |

allows DAMA to improve model alignment by preventing both overfitting on easy samples and underfitting on hard ones.

# B ADDITIONAL EXPERIMENTS

## B.1 ADVERSARIAL NOISE

We conduct additional experiments to explore the impact of iteration and step size in generating adversarial noise. As shown in Table 3, we report experimental results using `Qwen2-VL-7B` as the base model, and compare the performance of different iteraion and step size pairs. In the table, $+(i,j)$ means we construct offline dataset by adversarial attack on the image $i$ iterations with $\frac{j}{255}$ as each step size. The training setting is the same as the +*offline adv.* in our ablation study. Based on the results, we can see that the adversarial attack performs similarly on all benchmarks except for the MME and ObjectHal.

## B.2 CASE STUDY

In this section, we provide some detailed case studies of the model output for both online and offline datasets. In Fig. 5, we show an example to compare model responses on images with adversarial noise and random noise. The image with adversarial noise effectively triggers the prior biases of the LLM backbone. Furthermore, an example of model's responses to an online multiple-choice question is shown in Fig. 6. The response from `Qwen2-VL-7B` and `Qwen2.5-VL-7B` on online Yes/No data are shown in Fig. 7 and Fig. 8 respectively. On the online datasets, the model produces consistent outputs for some examples, while yielding different answers for others.

# C LIMITATION

As a preliminary exploration of using online data with verified reward to align LMMs, our work only study the alignment of image and text. This limits the ability of LMMs on other modality, such as video and audio. In the future, it is worth studying the effectiveness of using reinforcement learning with verified rewards to align LMMs on more modalities.

## USE OF LARGE LANGUAGE MODELS

Large language models were used exclusively for grammar checking and minor language refinement of the manuscript. They were not involved in generating ideas, designing methods, analyzing data, or interpreting results.

Figure 5: An example comparing model responses of the image with adversarial noise and random noise. The prior bias from LLM is marked in red.

Figure 6: An example of model responses on a multiple-choice question. Explanations are hidden for clarity. Each model generates 16 responses using random decoding. The correct responses from LMM are marked in green.

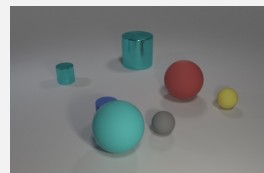

**Question: There is a gray rubber object; does it have the same shape as the metallic object left of the blue matte cylinder?**

**Correct Answer: No**

- *"No, the gray rubber object and the metallic object to its left in the image both have the same shape, which is a cylinder."*
- *"no"*
- *"No. The gray object is a small matte ball, which is the same shape as the other three balls in the image. It is not a cylinder like the metallic objects are."*
- *"No, the gray rubber object and the metallic object left of the blue matte cylinder are not the same shape. The gray rubber object is a sphere, while the metallic object is a cylinder.",*
- *"No, the gray rubber object in the picture has the same shape as the other spheres and balls in the image. It is also a ball or sphere."*
- *"No, the gray rubber object is the same shape as the blue matte cylinder, or any cylinders on the image. The gray rubber object appears to be a sphere due to its smooth surface and curvature."*
- *"No, the gray rubber object is also a sphere like the other objects in the image, while the metallic object left of the blue matte cylinder is a cylinder. cylinders and spheres are distinct shapes in geometry."*
- *"No, the gray rubber object does not have the same shape as the metallic object left of the blue matte cylinder. The gray object is a ball, while the metallic object is a cylinder."*
- *"No, the gray rubber object and the metallic object left of the blue matte cylinder have different properties and purposes. The gray rubber object is a small, smooth sphere that is often used in 3D modeling and animation to represent objects such as marbles, balls, or spheres. These objects are typically created in 3D modeling software and are not physically real."*
- *"No, the gray rubber object in the image is not the same shape as the metallic object left of the blue matte cylinder. The metallic object is a cylinder, while the gray object is a sphere. The differences in their shapes and forms make them distinct from each other."*
- *"No, the gray rubber object is a sphere, while the metallic object left of the blue matte cylinder is a cylinder."*
- *"Yes, the gray rubber object in the picture is a sphere, just like the other objects."*
- *"No, the gray rubber object does not have the same shape as the metallic cylinder on the left. The gray rubber object is a sphere, while the metallic cylinder is a cylinder."*
- *"No, the gray rubber object in the image is a small gray ball, which shares the same shape as the larger red ball, the small yellow ball, and another larger cyan ball. It is not the same shape as the metallic objects, which are either cylinders."*
- *"No, the gray rubber object does not have the same shape as the metallic object on the left. The metallic object is a cylinder, while the gray object is a sphere. Although they are both three-dimensional shapes, they differ in their geometry."*
- *"No, the gray rubber object and the metal object left of the blue matte cylinder are not the same shape. The gray rubber object is a small sphere, while the blue metal object is a cylinder."*

Figure 7: An example of responses from `Qwen2-VL-7B` on a Yes/No question. The model generates 16 responses using random decoding. The incorrect responses from the model are marked in red.

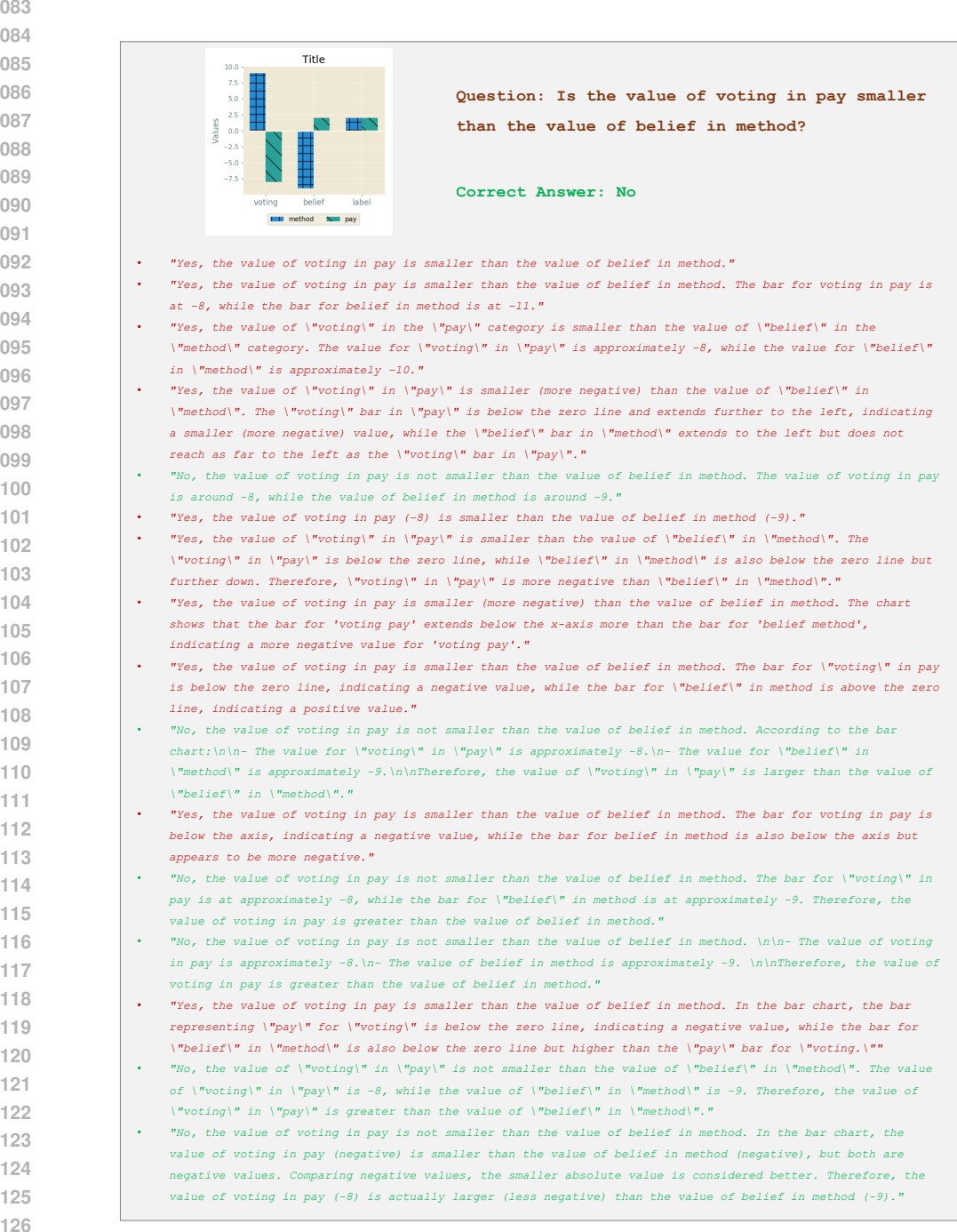

Figure 8: An example of responses from `Qwen2.5-VL-7B` on a Yes/No question. The model generates 16 responses using random decoding. The incorrect responses from the model are marked in red.

