# OpenReview forum: "Modality-Balancing Preference Optimization of Large Multimodal Models by Adversarial Negative Mining"
_ICLR.cc/2026/Conference — ICLR 2026 Conference Withdrawn Submission_

### Official Review · Reviewer_mFDa · 2025-10-29

**Soundness:** 3
**Presentation:** 3
**Contribution:** 3
**Rating:** 6
**Confidence:** 2

**Summary:**

This paper tackles one of the most persistent problems in multimodal model training: modality imbalance, where large multimodal models rely too heavily on their language backbone and ignore the visual evidence. The authors propose MBPO (Modality-Balancing Preference Optimization), a new framework that combines offline adversarial negative mining and online preference learning using verifiable rewards. The offline part generates “hard negatives” by adding adversarial perturbations to images, which push the model toward text-biased, less visually grounded answers. The online part collects dynamic data from closed-ended visual questions (like multiple choice or yes/no) and assigns verifiable rewards using GRPO. The two are combined in a hybrid training scheme. Experiments on multiple benchmarks (AI2D, MME, MMStar, MMVet, MMBench, MMHal-Bench) show consistent gains across both reasoning and hallucination metrics.

**Strengths:**

1) Modality imbalance is a serious and often-overlooked issue in multimodal alignment. The paper targets it directly with a concrete and well-designed optimization framework.

2) Using adversarial image perturbations to expose the model’s overreliance on text priors is an elegant way to produce meaningful “hard negatives.” The Image Information Gain (IIG) metric is a nice touch for quantifying visual grounding.

3) The experiments are extensive, covering both general vision-language understanding and hallucination reduction. The reported gains, especially in hallucination benchmarks, are clear and meaningful.

4) The paper provides solid evidence for each design choice. The visual comparisons between adversarial and random noise cases are intuitive and convincing.

**Weaknesses:**

1) The online dataset only uses around 2k closed-ended samples, which feels modest. It would be interesting to see whether scaling this part further would continue to improve results or plateau.

2) The IIG metric is intuitive but could benefit from more analysis. For instance, how correlated is IIG with human judgments of visual grounding?

**Questions:**

How sensitive is MBPO to the hyperparameters of adversarial noise?

---

### Official Review · Reviewer_Lc59 · 2025-10-31

**Soundness:** 2
**Presentation:** 1
**Contribution:** 2
**Rating:** 2
**Confidence:** 4

**Summary:**

This work introduces MBPO, Modality-Balancing Preference Optimization, a novel method combining offline preference pairs constructed with adversarial image perturbation with sampled online trajectories paired with verifiable rewards in a unified RL setting based on GRPO with an aim to enhance vision-language task performance and reduce hallucinations.

**Strengths:**

* The manuscript is well written and easy to follow with clear technical implementation details and the used data and models are openly available, facilitating reproducibility.
* The proposed method seems to produce consistent and non trivial improvements over relatively strong (i.e. already relatively well aligned) baselines (~+1% on Qwen2-VL-7B and ~+1.2% on Qwen2.5-VL-7B), despite minimal data requirements.

**Weaknesses:**

* Results in table 1 and 2 report “averages” which appear to actually be sums. Additionally, MME and MMHal* need to be normalized from 0-100 (instead of 0-2000 and 0-6) to allow for proper averaging.
* Most prior alignment work, including the works cited in this paper, report results for aligning LLaVA 1.5, which allows more direct comparison between methods In this work, the authors only report results on the more recent Qwen2 and Qwen2.5 base models. For direct comparison one must therefore rely on the author’s reproduction of prior work on this new base model. However, the reported reproductions seem to achieve no or only limited improvement over the Qwen baselines, which may suggest that either these previously published works are not generally effective on these newer base models or the reproductions may not be fully optimized. For example, with the adjusted averaging (norming MME and MMHal), for RLAIF-V the draft reports a -0.6% average regression over the Qwen2-VL-7B baseline and a modest +0.4% improvement over the Qwen2.5-VL-7B baseline. For CSR the numbers are -0.5% and +0.1%, etc. For a more direct comparison, perhaps MBPO could also be applied on LLaVA 1.5 for a direct comparison with published results?
* The offline data is produced by adversarially perturbing the input image to minimize the likelihood of the ground-truth response. The paper suggests that this would induce the model to rely on the “prior biases of the LLM backbone”, but doesn’t discuss why the resulting response would satisfy this criteria when e.g. an image that induces a refusal response may also satisfy the adversarial objective (though Figure 3 provides some empirical evidence).
* The paper reports hallucination via MMHal and CHAIR scores (without recall for CHAIR). Both these benchmarks exhibit some known weaknesses as discussed in [BDHS]. This provides a somewhat limited view on the resulting hallucination performance. Adding AMBER [AMBER], the vision enabled MMHalBench-V variant [BDHS], Pope [POPE], or even DeCapBench [PAINTING] may provide a more comprehensive picture.
* In the introduction, the paper claims that “Furthermore, the inherently offline nature of DPO—relying exclusively on pre-collected model responses—limits its ability to adapt to distributional shifts during training, thereby hindering optimization effectiveness”, though online as well as iterative DPO strategies have been successfully explored [OnlineDAP] [BDHS] [RLAIF-V].



[POVID] Zhou, Yiyang, et al. "Aligning modalities in vision large language models via preference fine-tuning." arXiv preprint arXiv:2402.11411 (2024).

[BDHS] Amirloo, Elmira, et al. "Understanding alignment in multimodal llms: A comprehensive study." arXiv preprint arXiv:2407.02477 (2024).

[PAI] Liu, Shi, Kecheng Zheng, and Wei Chen. "Paying more attention to image: A training-free method for alleviating hallucination in lvlms." European Conference on Computer Vision. Cham: Springer Nature Switzerland, 2024.

[RLAIF-V] Yu, Tianyu, et al. "Rlaif-v: Open-source ai feedback leads to super gpt-4v trustworthiness." Proceedings of the Computer Vision and Pattern Recognition Conference. 2025.

[POPE] Li, Yifan, et al. "Evaluating object hallucination in large vision-language models." arXiv preprint arXiv:2305.10355 (2023).

[PAINTING] Ye, Qinghao, et al. "Painting with words: Elevating detailed image captioning with benchmark and alignment learning." arXiv preprint arXiv:2503.07906 (2025).

[OnlineDAP] Guo, Shangmin, et al. "Direct language model alignment from online ai feedback." arXiv preprint arXiv:2402.04792 (2024).

[AMBER] Wang, Junyang, et al. "Amber: An llm-free multi-dimensional benchmark for mllms hallucination evaluation." arXiv preprint arXiv:2311.07397 (2023).

**Questions:**

* In this work, the data for offline preference construction is sampled to contain samples exhibiting high IIG. What motivated this choice? To me, the choice is not clearly intuitive. Some prior works (also cited in this paper) describe a relationship with lack of attention to visual tokens and visual hallucinations [Povid] [PAI] [BDHS], which would intuitive translate to low IIG scores suggesting that such sampling may yield a set where the base model may be less likely to hallucinate on to begin with? On the other hand, sampling for IIG may yield samples where the image content is more necessary to answer the questions yielding samples that could be informative for training. Was any further study done on the impact of the choice in the paper?
* In section 4.1 the work describes that a reward of 2 is given “to the chosen response”. Does this mean reward is only assigned with the chosen response is correctly and fully reproduced (i.e. full string matching)? Unless this data was seen during SFT, would this not be difficult for the model?
* In the reproduction of POVID, was the image distortion objective reproduced as described in the original paper? [POVID]
* In the reproduction of RLAIF-V, was the same iterative training method followed, which collects online feedback? (see section 2.3 in [RLAIF-V])
* The authors compare the proposed adversarial noise injection with random noise as proposed in [POVID], however, as mentioned in that paper and later also studied in [BDHS] injecting such noise needs some calibration in order to not diverge too much or too little from the original context. The example in figure 5, which shows no errors in the noised responses when random noise is used, may indicate that the noising hyper parameters are not chosen optimally, potentially under-reporting this baseline performance?



[POVID] Zhou, Yiyang, et al. "Aligning modalities in vision large language models via preference fine-tuning." arXiv preprint arXiv:2402.11411 (2024).

[BDHS] Amirloo, Elmira, et al. "Understanding alignment in multimodal llms: A comprehensive study." arXiv preprint arXiv:2407.02477 (2024).

[PAI] Liu, Shi, Kecheng Zheng, and Wei Chen. "Paying more attention to image: A training-free method for alleviating hallucination in lvlms." European Conference on Computer Vision. Cham: Springer Nature Switzerland, 2024.

[RLAIF-V] Yu, Tianyu, et al. "Rlaif-v: Open-source ai feedback leads to super gpt-4v trustworthiness." Proceedings of the Computer Vision and Pattern Recognition Conference. 2025.

[POPE] Li, Yifan, et al. "Evaluating object hallucination in large vision-language models." arXiv preprint arXiv:2305.10355 (2023).

[PAINTING] Ye, Qinghao, et al. "Painting with words: Elevating detailed image captioning with benchmark and alignment learning." arXiv preprint arXiv:2503.07906 (2025).

[OnlineDAP] Guo, Shangmin, et al. "Direct language model alignment from online ai feedback." arXiv preprint arXiv:2402.04792 (2024).

---

### Official Review · Reviewer_t7GD · 2025-11-01

**Soundness:** 1
**Presentation:** 1
**Contribution:** 1
**Rating:** 2
**Confidence:** 4

**Summary:**

This paper introduces MBPO (Modality-Balancing Preference Optimization) for aligning Large Multimodal Models (LMMs) under modality imbalance - the tendency to over-weight language priors relative to visual evidence. MBPO combines (i) an offline preference set built via *adversarially mined negatives* that explicitly suppress image information usage, and (ii) an online phase that applies GRPO on close-ended, verifiable prompts to provide stable rewards with format/length control. Experiments (via LMMS-Eval) on hallucination suites indicate improved general VL capability and reduced hallucination compared with strong preference-learning baselines. The paper argues MBPO addresses the root cause - unbalanced reliance on text - rather than only post-hoc mitigation.

**Strengths:**

1. **Key Idea is  not hard to understand.** The *adversarial-negative mining* pipeline is used to construct preference pairs that penalize language-only shortcuts, while GRPO supplies online, distribution-adaptive, verifiable signals that are robust to reward hacking on free-form answers.
2. **Positioning vs. preference learning.** The paper rightly notes that **DPO** can under-use images in multimodal settings, and that mDPO explicitly tackles “unconditional preference” effects.

**Weaknesses:**

1. **Unclear Motivation & Presentation.** The motivation is wired to me. The claim that current LMMs can neglect visual information is acceptable somehow, but what is the point of introducing GRPO? Why do we need online-offline mixing mode for preference optimization given your claim?
   Also, the writing and presentation in implementation details in the whole paper is not clear.
2. **Attribution of gains across components is under-specified.** The paper should isolate and quantify the marginal utility of (a) adversarial-negative mining, (b) IIG-style filtering, (c) online GRPO over close-ended prompts, and (d) the format/length reward. Especially, why do we need adversarial attacks for mining the negative samples? The grounding details lack, especially for equation (6) which demonstrates the gradient descent process.
3. **Problem in evaluation.** Parts of MM-Vet/MMHal pipelines leverage LLM-as-judge scoring, which can not only be prompt- and model-sensitive, but also contain certain bias during judging.
4. **Head-to-head with conditional preference methods.** Since mDPO conditions explicitly on the visual signal to avoid image-agnostic learning, a strictly controlled comparison (identical backbone, data budget, prompts) is essential to establish MBPO’s advantage on the same failure mode.
5. **Grounding diagnostics.** Claims about “increased visual usage” would be more convincing with patch-/token-level attribution analyses (e.g., cross-modal attention concordance, counterfactual masking) or faithfulness metrics beyond CHAIR.
6. **No conclusion?** There seems not have the section for conclusion.
7. **Generalization across backbones/scales.** Most results center on a single family (e.g., Qwen-VL, 7B). Adding smaller/larger or LLaVA-style backbones would improve external validity.
8. **Non-substantial (and wired) evaluation.** The evaluation frameworks are not clear, especially the evaluation metrics for each dataset. Please specify. Also, please consider more benchmarks such as HallusionBench[1] and MMStar[2] for comprehensive (both discriminative and generative) hallucination evaluation settings.

References:

[1] Guan et al., "HallusionBench: An Advanced Diagnostic Suite for Entangled Language Hallucination and Visual Illusion in Large Vision-Language Models" In CVPR 2024.

[2] Chen et al., "Are we on the right way for evaluating large vision-language models?" In NeurIPS 2024.

**Questions:**

1. **Component attribution.** Can you provide a factorial ablation for mining vs. IIG vs. online GRPO vs. format reward, with confidence intervals on MM-Vet/MME and CHAIR/MMHal?
2. **Evaluator robustness.** How stable are rankings under different judge prompts or alternative evaluators? Any small-scale human audit?
3. **Generalization.** Results on additional backbones/scales and on sequence-understanding variants would help validate scope.
4. **Failure modes.** When do adversarial negatives over-penalize benign visual variations (lighting/cropping) and reduce robustness?

---

### Official Review · Reviewer_M7oV · 2025-11-06

**Soundness:** 3
**Presentation:** 2
**Contribution:** 3
**Rating:** 2
**Confidence:** 4

**Summary:**

This paper proposes Modality-Balancing Preference Optimization (MBPO), a hybrid offline–online preference learning framework addressing the issue of modality imbalance in large multimodal models (LMMs). The method generates adversarially perturbed “hard negative” samples by modifying input images, which amplifies language prior bias and encourages the model to rely more on visual evidence. Complementary online preference data are collected from closed-ended, verifiable tasks, and both offline and online data streams are jointly optimized through Group Relative Policy Optimization (GRPO). Experiments show that MBPO improves performance on a range of benchmarks, effectively reducing hallucinations and enhancing visual grounding.

**Strengths:**

1.Clear Motivation and Problem Definition: The paper convincingly identifies modality imbalance—where language priors overshadow visual information—as a critical limitation in current LMMs. The introduction of the Image Information Gain (IIG) metric to quantify visual contribution is conceptually elegant and practically useful.
2. Innovative Adversarial Negative Mining: Instead of random perturbations, MBPO employs adversarial attacks to construct meaningful hard negatives. This strategy effectively exposes models’ overreliance on linguistic priors, as shown in Figures 1, 2, and 5.

**Weaknesses:**

1. Ablation and Parameter Analysis Limitations: While Table 2 and Table 3 do present several ablations, some aspects remain under-explored. For example, it’s unclear whether the observed gains predominantly stem from data curation (selecting better/harder negatives), the adversarial perturbation technique, or the hybrid GRPO objective. More fine-grained ablations, including IIG score cutoff sensitivity and adversarial attack hyperparameters, would help clarify these contributions and potential confounders
2. Potentially Overstated Claims on Hallucination Mitigation: The improvements in hallucination rates (e.g., in Table 1, as measured by CHAIR_S and CHAIR_I) are promising but in some cases modest, and it’s not always clear that reductions are statistically significant across all benchmarks.

**Questions:**

1. What are the consequences of using the model’s own likelihood for both generating adversarial examples and computing IIG; could this lead to confirmation bias or spurious improvements?
2. How do performance gains in hallucination metrics (e.g., CHAIR_S, CHAIR_I) translate into actual user-facing improvements?

---

### Note · Authors · 2025-11-13

I have read and agree with the venue's withdrawal policy on behalf of myself and my co-authors.